# The Protective Effect against Lung Injury of Phytosome Containing the Extract of Purple Waxy Corn Tassel in an Animal Model of PM2.5-Induced Lung Inflammation

**DOI:** 10.3390/foods13203258

**Published:** 2024-10-13

**Authors:** Nut Palachai, Wipawee Thukham-mee, Jintanaporn Wattanathorn

**Affiliations:** 1Faculty of Medicine, Mahasarakham University, Mahasarakham 44000, Thailand; nut.p@msu.ac.th; 2Integrative Complementary Alternative Medicine Research and Development Center in the Research Institute for Human High Performance and Health Promotion, Khon Kaen University, Khon Kaen 40002, Thailand; meewep@gmail.com; 3Department of Physiology, Faculty of Medicine, Khon Kaen University, Khon Kaen 40002, Thailand

**Keywords:** inflammation, respiratory diseases, phytosome, purple waxy corn, tassel, anthocyanins

## Abstract

Lung inflammation caused by fine particulate matter (PM), particularly PM2.5, poses a significant public health challenge, with oxidative stress and inflammation playing central roles in its pathophysiology. This study evaluates the protective effects of phytosome-encapsulated extract of purple waxy corn tassel (PPT) against PM2.5-induced lung inflammation. Male Wistar rats received PPT at doses of 100, 200, and 400 mg/kg BW for 21 days prior to exposure and continued to receive the same doses for 27 days during PM2.5 exposure. Significant reductions in inflammatory markers, including cyclooxygenase-2 (COX-II), various interleukins (IL-1β, IL-6, IL-8), tumor necrosis factor-alpha (TNF-α), and nuclear factor kappa B (NF-κB), were observed, indicating that PPT effectively regulates the inflammatory response. Additionally, PPT improved oxidative stress markers by reducing malondialdehyde (MDA) levels and enhancing antioxidant enzyme activities such as superoxide dismutase (SOD), catalase (CAT), and glutathione peroxidase (GSH-Px), thereby restoring lung antioxidant defenses. Notably, the study revealed that PPT modulates epigenetic mechanisms, as evidenced by decreased histone deacetylase (HDAC) activity and upregulation of sirtuins in lung tissue. These epigenetic modifications likely contribute to the reduction in inflammation and oxidative stress, suggesting a multifaceted protective role of PPT that involves both direct biochemical pathways and epigenetic regulation. The interplay between reduced inflammatory signaling, enhanced antioxidant capacity, and epigenetic modulation underscores PPT’s potential as a therapeutic agent for managing respiratory inflammation-related diseases and its promise for the development of future functional food products.

## 1. Introduction

The rising prevalence of respiratory diseases has become a critical public health concern globally, particularly in many Asian countries, where urbanization and industrialization have intensified exposure to environmental pollutants. Recent epidemiological studies indicate that respiratory conditions such as asthma, chronic obstructive pulmonary disease (COPD), and lung cancer are not only increasing in incidence but are also becoming more severe, leading to significant morbidity and mortality [1,2,3]. In this context, fine particulate matter with an aerodynamic diameter of less than 2.5 μm (PM2.5) has emerged as a major environmental hazard. Composed of a complex mixture of solid and liquid particles, PM2.5 penetrates deep into the respiratory tract, eliciting a range of pathological responses that contribute to the development and exacerbation of various respiratory diseases [4].

The mechanisms by which PM2.5 induces respiratory pathology are multifaceted, with oxidative stress and inflammation playing pivotal roles. Exposure to PM2.5 has been shown to increase the generation of reactive oxygen species (ROS), leading to oxidative damage of cellular components and triggering inflammatory pathways [5]. This cascade of events results in the activation of pro-inflammatory mediators, which can perpetuate lung injury and contribute to chronic respiratory conditions. Despite the growing evidence linking PM2.5 exposure to deteriorating respiratory health, current therapeutic options are largely inadequate, with available treatments focusing mainly on symptomatic relief, rather than addressing the underlying oxidative stress and inflammation that drive the disease process [6,7]. This limitation underscores the urgent need for innovative preventive strategies and therapeutic interventions that effectively target these pathophysiological processes.

In the search for alternative therapeutic approaches, phytochemicals have gained attention for their potential health benefits. Compounds such as polyphenols, flavonoids, and anthocyanins exhibit notable antioxidant and anti-inflammatory properties, suggesting they may provide protective effects against oxidative stress-induced damage [8,9,10]. Although numerous studies have focused on purple waxy corn, none have specifically examined the tassel part as a functional food. This represents a significant research gap, making this the first study to report on the new biomedical activity of purple waxy corn tassel as a functional food [11,12]. However, a significant barrier to the clinical application of these phytochemicals is their poor bioavailability and stability, which can severely limit their efficacy [13,14,15]. Additionally, this study is the first to explore phytosome encapsulation technology as a formulation to enhance the activity and stability of purple waxy corn tassel.

Recent advancements in phytosome encapsulation technology present a viable solution to these challenges, enhancing the bioavailability and stability of phytochemicals and improving their therapeutic potential [10,16]. This study hypothesizes that phytosome-encapsulated extracts from purple waxy corn tassels can effectively attenuate respiratory inflammation in an experimental model exposed to PM2.5. To investigate this hypothesis, we assess the effects of the phytosome formulation on lung histopathology, inflammatory markers (COX-II, IL-1β, IL-6, IL-8, TNF-α, NF-κB), and oxidative stress markers (MDA, SOD, CAT, GSH-Px). Additionally, we aim to explore the potential epigenetic modifications induced by this intervention, specifically focusing on HDAC activity and sirtuins, which play critical roles in regulating inflammation and cellular stress responses. By addressing these key aspects, this research aims to elucidate the protective mechanisms of phytosome-encapsulated purple waxy corn tassel extract and contribute to the growing body of knowledge on functional foods as potential therapeutic options for managing respiratory diseases linked to environmental pollutants. The findings from this study could lay the groundwork for future applications in food science, highlighting the importance of incorporating bioactive compounds from natural sources into strategies aimed at mitigating the adverse health effects of air pollution.

## 2. Materials and Methods

### 2.1. PPT Preparation

The purple waxy corn tassel utilized in this study was sourced from the Faculty of Agriculture at Khon Kaen University, Thailand. A voucher specimen (KKU No.25979) was archived at the Research Institute for Human High Performance and Health Promotion, Khon Kaen University, Thailand. The tassels were thoroughly cleaned and dried in a Memmert oven (Memmert GmbH, Elkhart, IN, USA) at 60 °C for 72 h. Once dried, they were ground into a fine powder. To prepare the extract, the powdered tassel was subjected to a 50% hydroalcoholic extraction process using maceration. The mixture was then centrifuged at 3000 rpm for 10 min and filtered through Whatman No. 1 filter paper. The resulting filtrate was concentrated using a rotary evaporator and subsequently freeze-dried using a Labconco freeze dryer (Labconco Corporation, Kansas City, MO, USA).

The phytosome preparation of purple waxy corn tassel (PPT) followed the method established by Palachai et al. [10]. Briefly, the 50% hydroalcoholic extract of the tassel was dissolved in 50% ethanol, while phosphatidylcholine was dissolved in dichloromethane. The two solutions were combined and stirred at 25 °C for 8 h. Following this, the ethanol and dichloromethane were removed using a rotary evaporator at 45 °C for 3 h. The final product was dried using a spray dryer (BUCHI Mini Spray Dryer B-290, BÜCHI Labortechnik AG, Flawil, Switzerland). The dried PPT was stored in a desiccator with silica gel at 4 °C for preservation.

### 2.2. Assessment of the Fingerprint Chromatogram

The fingerprint chromatogram was analyzed through high-performance liquid chromatography (HPLC). A Waters^®^ system, equipped with a Waters^®^ 2998 photodiode array detector, was used for this purpose. Chromatographic separation was achieved using a Purospher^®^ STAR C-18 endcapped column (5 μm), LiChroCART^®^ 250-4.6, and HPLC Cartridge, Sorbet Lot No. HX255346 (Merck KGaA, Darmstadt, HE, Germany). The mobile phase consisted of 100% methanol (solvent A) (Fisher Scientific, USA) and 2.5% acetic acid (solvent B) (Fisher Scientific, USA) in deionized water, with a gradient elution. The gradient was programmed as follows: 70% solvent A from 0 to 17 min, 100% solvent A from 18 to 20 min, and 10% solvent A from 20.5 to 25 min. The flow rate was maintained at 1.0 mL/min. Prior to injection, the sample was filtered using a 0.45 µm Millipore filter (Merck KGaA, Darmstadt, HE, Germany), and a 20 µL aliquot was directly injected into the system. Detection of the chromatogram was carried out at 280 nm using a UV detector, and EmpowerTM3, Version 3.4 software was employed for data analysis [10].

### 2.3. Assessment of Total Phenolic Compounds, Total Flavonoids, and Total Anthocyanins Contents

The total phenolic content in the sample was measured using the Folin–Ciocalteu colorimetric method with a microplate reader (iMark™ Microplate Absorbance Reader). A freshly prepared reagent composed of 158 µL of distilled water and 20 µL of 50% v/v Folin–Ciocalteu reagent (Sigma-Aldrich, St. Louis, MO, USA) was added to 20 µL of the sample, followed by an 8 min incubation. Afterward, 30 µL of 20% Na_2_CO_3_ (Sigma-Aldrich, USA) was introduced, and the mixture was incubated at 25 °C in the dark for 2 h. Absorbance was then measured at 765 nm, and the results were expressed as mg gallic acid equivalents (GAE) per mg of extract. A standard calibration curve was generated using various concentrations of gallic acid (Sigma-Aldrich, USA) [17].

For the total flavonoid content, the aluminum chloride colorimetric method was employed. In brief, 100 µL of the sample at different concentrations was mixed with 100 µL of 2% methanolic aluminum chloride (Sigma-Aldrich, USA) and incubated in the dark at 25 °C for 30 min. Following incubation, the absorbance was measured at 415 nm against a blank. A standard calibration curve was created using different concentrations of quercetin (Sigma-Aldrich, USA), and the results were reported as mg quercetin equivalents per mg of extract [18].

The total anthocyanin content was assessed using the pH-differential method. Two dilutions of the sample were prepared, one with sodium acetate buffer (pH 4.5) and another with potassium chloride buffer (pH 1.0), based on a predetermined dilution factor. The samples were incubated at 25 °C for 20 min before measuring absorbance at 510 nm and 700 nm. Total anthocyanin content was calculated as mg of cyanidin-3-glucoside equivalents per mg of extract using the molar extinction coefficient (ɛ) of cyanidin-3-O-glucoside (26,900 L mol^−1^ cm^−1^) and a molar weight of 449.2 g mol^−1^ [19].

### 2.4. Assessment of Biological Activities

The 1,1-diphenyl-2-picrylhydrazyl radical (DPPH) assay was employed to assess the sample’s free radical-scavenging capacity, using the stable DPPH. A 0.1 mM DPPH solution in methanol was prepared, and 2 mL of this solution was mixed with 0.3 mL of the sample at various concentrations (1–100 mg/mL). The mixture was incubated for 30 min at 25 °C, after which the absorbance was measured at 517 nm using a microplate reader (iMark™ Microplate Absorbance Reader), with a blank containing no sample. L-ascorbic acid served as the positive control, and results were expressed as EC50, representing the concentration (in µg/mL) needed to inhibit 50% of the radical activity [20].

The FRAP (Ferric Reducing Antioxidant Power) assay evaluated the sample’s ability to reduce ferric tripyridyltriazine (Fe^3+^-TPTZ) to ferrous tripyridyltriazine (Fe^2+^-TPTZ). The FRAP reagent was prepared by combining 300 mM acetate buffer, 10 mM TPTZ, and 20 mM ferric chloride (FeCl_3_) solutions in a 10:1:1 ratio. A mixture of 190 µL FRAP reagent and 10 µL of the sample was incubated at 37 °C for 10 min, and absorbance was measured at 593 nm. Ascorbic acid served as the positive control, with the results expressed as EC50 [21].

The 2,2′-azino-bis(3-ethylbenzothiazoline-6-sulfonic acid) (ABTS) assay was utilized to further assess the sample’s free radical-scavenging activity. The ABTS•^+^ solution was prepared by mixing 7 mM ABTS and 2.45 mM potassium persulfate (K_2_S_2_O_8_) in a 2:3 ratio. To perform the assay, 30 µL of the sample at varying concentrations was combined with 120 µL of distilled water and 30 µL of ethanol. The resulting solution was reacted with 3 mL of the ABTS•^+^ solution, and the absorbance was measured at 734 nm using a spectrophotometer (Pharmacia LKB-Biochrom, Uppsala, Uppsala, Sweden). Trolox was used as the standard, with results again expressed as EC50 [22].

The inhibition of COX-II was evaluated using a colorimetric COX-II inhibitor screening assay kit (Cayman Chemical, Ann Arbor, MI, USA). The assay followed the manufacturer’s protocol, with the COX-II working solution prepared by dissolving COX-II in 100 mM Tris-HCl buffer (pH 8.0) at a ratio of 1:100. The reaction mixture contained 150 µL of assay buffer, 10 µL of sample, 10 µL of heme, 10 µL of COX-II working solution, 20 µL of 10 µM TMPD (N,N,N′,N′-Tetramethyl-p-phenylenediamine dihydrochloride), and 20 µL of 100 µM arachidonic acid. This mixture was added to a 96-well microplate and incubated at 25 °C for 30 min. Absorbance at 590 nm was recorded, with results expressed as EC50. Indomethacin was used as a reference standard [23].

### 2.5. Experimental Protocol

Male Wistar rats (180–220 g, 8 weeks old) were sourced from Nomura Siam International Co., Ltd. The rats were maintained under standard laboratory conditions with free access to food and water, housed six per cage in metal enclosures. The temperature was controlled at 23 ± 2 °C with a 12:12-h light-dark cycle. All procedures were approved by the Institutional Animal Ethics Committee of Khon Kaen University (Approval No. IACUC-KKU 11/64). Following a 1-week acclimatization period, the rats were randomly assigned into six groups (*n* = 6), according to the necessity to minimize animal use in line with the 3Rs (Replacement, Reduction, Refinement), the demonstrated superior bioactivity of PPT in prior in vitro studies, and the aim to elucidate the specific effects and mechanisms of the phytosome formulation as a therapeutic option for PM2.5-induced respiratory inflammation. The groups were:

Group I (naïve control): Rats in this group received a standard diet and vehicle treatment.

Group II (respiratory inflammation + vehicle): Rats in this group were given a vehicle orally for 21 days prior to and during the 27-day PM2.5 exposure period.

Group III (respiratory inflammation + prednisolone): Rats received 1 mg/kg BW of prednisolone (positive control) orally for 21 days prior to and during the 27-day PM2.5 exposure.

Groups IV–VI (respiratory inflammation + PPT100, PPT200, and PPT400): Rats received PPT at doses of 100, 200, or 400 mg/kg BW orally for 21 days prior to and during the 27-day PM2.5 exposure.

The PM2.5-induced lung inflammation model was adapted from the versatile aerosol concentration enrichment system by Sioutas et al. [24]. Briefly, the rats were placed in a clear plastic chamber with an air filter (30.5 × 52.5 × 17 cm). PM2.5 (Sigma-Aldrich, St. Louis, MO, USA; product number NIST1648A) suspension was administered via a pressure-cycled ventilator (CW-SAR-830/AP, World Precision Instruments, Sarasota, FL, USA). The concentration of particulate matter was maintained at a minimum of 100 µg/m^3^, monitored using a particle counter (HT-9600/HTI, Tübingen, Germany). The rats were exposed to PM2.5 at 10× ambient concentrations for 3 h per day, 5 days per week, over a 21-day period.

The animals were administered their assigned substances once daily throughout the 21-day pre-exposure and 27-day PM2.5 exposure periods. Daily monitoring of food and water intake, along with body weight, was conducted. At the conclusion of the study, lung tissue was analyzed for inflammatory markers, including histomorphological changes, COX-II activity, and levels of IL-1β, IL-6, IL-8, TNF-α, and NF-κB. Additionally, oxidative stress markers, as well as sirtuin expression and HDAC activity, were assessed. A schematic diagram of the experimental design is shown in Figure 1.

### 2.6. Assessment of Histopathological Abnormalities in Lung Tissue

Following euthanasia, lung tissues were excised and placed in 10% formalin (Sigma-Aldrich, USA) for fixation over a period of 72 h. The tissue samples were then cryosectioned at 10 µm thickness using a Thermo Scientific™ HM 525 Cryostat (Thermo Scientific, Waltham, MA, USA). The sections were mounted on slides pre-coated with a 0.3% aqueous gelatin solution containing 0.05% aluminum potassium sulfate (Sigma-Aldrich, USA). Afterward, the samples underwent hematoxylin and eosin (H&E) staining as described in a previous protocol [10].

Histopathological abnormalities were examined in three randomly selected fields per sample under a 40× magnification using an Olympus BH-2 light microscope (Olympus Corporation, Tokyo, Japan) with the PixeLINK PL-A6xx Capture (PixeLINK, Ottawa, ON, Canada) and IT tool program (PixeLINK Capture Software Version 4.1). The pathological changes in lung tissue were scored on a scale of 0 to 5, as described in earlier studies: 0 represented normal tissue; 1 indicated minimal inflammatory changes; 2 corresponded to no significant lung architecture damage; 3 involved alveolar septal thickening; 4 indicated the presence of nodules or pneumonitis regions distorting normal architecture; and 5 reflected total obliteration of the field [25,26,27,28].

### 2.7. Assessment of Oxidative Stress Status in Lung Tissue

Lung tissues were homogenized in 0.1 M potassium phosphate buffer (pH 7.4) at a dilution of 10 mg tissue per 50 µL of buffer. Protein concentrations in the homogenates were determined using a Thermo Scientific NanoDrop 2000c spectrophotometer (Thermo Fisher Scientific, Waltham, MA, USA) by measuring optical density at 280 nm.

The activity of SOD was evaluated using the method of Sun et al. [29]. The reaction mixture included 0.2 M phosphate buffer (KH_2_PO_4_, pH 7.8), 0.01 M EDTA, 15 µM cytochrome C, and 0.5 mM xanthine (pH 7.4), all from Sigma-Aldrich, USA. Lung homogenates (20 µL) were added to the mixture (200 µL) along with 20 µL of xanthine oxidase (0.90 mU/mL). The optical density was measured at 415 nm. SOD standards (0–25 U/mL) were used for reference, and results were expressed as units per mg protein.

CAT activity was measured by its ability to decompose hydrogen peroxide. Lung homogenates (10 µL) were mixed with 50 µL of 30 mM hydrogen peroxide (in 50 mM phosphate buffer, pH 7.0), 25 µL of 4 M H_2_SO_4_, and 150 µL of 5 mM KMnO_4_. The optical density was measured at 490 nm. CAT standards (10–100 U/mL) were used, and results were expressed as units per mg protein [30].

GSH-Px activity was assessed by mixing 20 µL of lung homogenate with 10 µL of 1 mM dithiothreitol, 10 mM monosodium phosphate, 1 mM sodium azide, 100 µL of 40 mM phosphate buffer (pH 7.0), 50 mM glutathione, and 30% hydrogen peroxide. The mixture was incubated at 25 °C for 10 min, after which 10 µL of 10 mM DTNB was added. Optical density was measured at 412 nm. GSH-Px standards (1–5 U/mL) were used as references, with results expressed as units per mg protein [31].

Lipid peroxidation was determined by measuring MDA via a thiobarbituric acid reaction. An aliquot of 50 µL lung homogenate was mixed with 50 µL of 8.1% sodium dodecyl sulfate, 375 µL of 0.8% thiobarbituric acid, 375 µL of 20% acetic acid, and 150 µL of distilled water. The mixture was heated at 95 °C for 60 min, then cooled and combined with 1250 µL of n-butanol and pyridine (15:1 ratio) and 250 µL of distilled water. After centrifugation at 4000 rpm for 10 min, the upper layer was collected, and optical density was measured at 532 nm. TMP (1,1,3,3-tetramethoxypropane) standards (0–15 µM) were used, and MDA levels were expressed as ng per mg protein [32].

### 2.8. Assessment of COX-II Activity in Lung Tissue

COX-II activity was quantified using an ELISA kit (COX Activity Assay Kit, item No. 760151, Cayman Chemical, Ann Arbor, MI, USA) according to the manufacturer’s protocol. Lung tissue samples were homogenized in 0.1 M Tris-HCl buffer (pH 7.8) containing 1 mM EDTA. The homogenate was centrifuged at 10,000× *g* for 15 min at 4 °C. After centrifugation, 40 µL of the supernatant was combined with 120 µL of assay buffer and 10 µL of hemin. This mixture was agitated briefly and incubated for 5 min at 25 °C.

Subsequently, 20 µL of the colorimetric substrate and 20 µL of arachidonic acid were added to the solution, followed by another 5 min incubation at 25 °C. Absorbance was measured at 590 nm. COX-II activity was calculated using the following equation:Total COX-II activity = ((ΔA590/5 min)/(0.00826 µM^−1^)) × (0.21 mL/0.04 mL)/2

### 2.9. Assessment of Histone Deacetylase Activity in Lung Tissue

HDAC activity was evaluated using a fluorometric HDAC activity assay kit (ab156064, Abcam, Cambridge, MA, USA) following the manufacturer’s instructions. Initially, 50 µL of lung tissue homogenate was combined with deionized distilled water (ddH_2_O), HDAC assay buffer, and 0.2 mM substrate peptide. For the inhibitor control, 10 µM of trichostatin A (an HDAC inhibitor, Abcam, Cambridge, MA, USA) was added.

To initiate the reactions, 5 µL of HDAC (used as a positive control) was added to each well and incubated for 20 min at 25 °C. After incubation, 20 µL of stop solution and 5 µL of developer were added and further incubated for 10 min at 25 °C. The optical density was then measured at wavelengths of 355/460 nm using a Varioskan LUX plate reader (Thermo Fisher Scientific, Waltham, MA, USA). HDAC activity was calculated using the formula:HDAC activity = enzyme sample assay − no enzyme control assay

### 2.10. Western Blotting Analysis

Lung tissue was isolated, homogenized, and lysed in 1/5 (*w*/*v*) RIPA (radioimmunoprecipitation assay) buffer (Cell Signaling Technology, Danvers, MA, USA), which contained 20 mM Tris-HCl (pH 7.5), 150 mM NaCl, 1 mM Na_2_EDTA, 1 mM EGTA, 1% NP-40, 1% sodium deoxycholate, 2.5 mM sodium pyrophosphate, 1 mM beta-glycerophosphate, 1 mM Na_3_VO_4_, 1 µg/mL leupeptin, and 1 mM phenylmethanesulfonyl fluoride (PMSF) (Cell Signaling Technology, USA). The homogenate was centrifuged at 12,000× *g* for 10 min at 4 °C, and the supernatant from the middle layer was collected. Protein concentration was measured using a Thermo Scientific NanoDrop 2000c spectrophotometer (Thermo Fisher Scientific, Wilmington, DE, USA).

For Western blotting, 80 µg of protein lysates were adjusted to the appropriate concentration with Tris-Glycine SDS-PAGE loading buffer and heated at 95 °C for 10 min. The proteins were separated by sodium dodecyl sulfate-polyacrylamide gel electrophoresis (SDS-PAGE), loading 20 µL of the sample onto the gel. The separated bands were transferred to a nitrocellulose membrane, washed with 0.05% TBS-T, and incubated in blocking buffer (5% skim milk in 0.1% TBS-T) at 25 °C for 1 h (Appendix A).

Following the blocking period, the membrane was incubated overnight at 4 °C with primary antibodies: anti-sirtuins (Cell Signaling Technology, USA; dilution 1:500), anti-IL-1β (Abcam, Cambridge, UK; dilution 1:500), anti-IL-6 (Cell Signaling Technology, USA; dilution 1:500), anti-IL-8 (Cell Signaling Technology, USA; dilution 1:500), anti-TNF-α (Cell Signaling Technology, USA; dilution 1:500), anti-NF-κB (Cell Signaling Technology, USA; dilution 1:500), and anti-β-actin (Cell Signaling Technology, USA; dilution 1:500). After incubation, the membrane was rinsed with TBS-T (0.05%), then incubated with anti-rabbit IgG, HRP-linked antibody (Cell Signaling Technology, USA; dilution 1:2000) at 25 °C for 1 h.

Bands were visualized and quantified using ECL detection systems (GE Healthcare, Little Chalfont, Buckinghamshire, UK) and an LAS-4000 luminescent image analyzer (GE Healthcare). Band intensities were analyzed using ImageQuant TL v.7.0 image analysis software (GE Healthcare) and normalized to anti-β-actin. Data were expressed as relative density compared to the naïve control group [33].

### 2.11. Statistical Analysis

Data are presented as mean ± standard error of the mean (SEM). Statistical significance was assessed using one-way analysis of variance (ANOVA), followed by the post hoc Tukey test. For comparisons between two groups, the Student’s *t*-test was employed. A *p*-value of less than 0.05 was considered statistically significant. All statistical analyses were conducted using SPSS version 21.0 (IBM Corp, Armonk, NY, USA). Released 2012. IBM SPSS Statistics for Windows).

## 3. Results

### 3.1. Phenolic Compositions and Biological Activities

Figure 2 presents the chromatogram of anthocyanins, showing cyanidin-3-glucoside as the main compound in both the extract and the PPT. The peak symmetry factor for cyanidin-3-glucoside was 1.0, the theoretical plate number was 7371, and the retention time was 48.712 min, indicating an acceptable level of peak symmetry and chromatographic efficiency.

To further validate the presence of this compound, we analyzed the UV absorption spectrum obtained from the HPLC analysis. The UV spectrum exhibited significant peaks in the range of 450–550 nm, with a maximum at around 520 nm, characteristic of cyanidin-3-glucoside due to its aromatic structure and conjugated double bonds. We compared the obtained UV spectrum with reference spectra from established databases and the literature, confirming the identity of cyanidin-3-glucoside based on matching peaks [34]. Notably, the peak at 516 nm demonstrated pronounced absorbance, indicating a high concentration of cyanidin-3-glucoside in our phytosome-encapsulated extract.

Table 1 provides the content of phenolic compounds, flavonoids, and anthocyanins. The analysis indicates that the 50% hydroalcoholic extract of purple waxy corn tassel contains 40.257 ± 0.024 mg GAE/mg extract of phenolic compounds, 5.938 ± 0.006 mg quercetin/mg extract of flavonoids, and 8.921 ± 0.056 mg C3G/mg extract of anthocyanins. In comparison, the phytosome formulation (PPT) contains 41.819 ± 0.259 mg GAE/mg extract of phenolic compounds, 5.749 ± 0.019 mg quercetin/mg extract of flavonoids, and 8.780 ± 0.028 mg C3G/mg extract of anthocyanins.

The standard curves were developed to ensure accuracy in the quantification. For phenolic compounds, the curve was generated using gallic acid, ranging from 10 to 1000 µg/mL (y = 0.0001x − 0.003, R^2^ = 0.999). For flavonoids, the standard curve was based on quercetin, ranging from 10 to 1000 µg/mL (y = 0.0003x − 0.3982, R^2^ = 0.997). Similarly, the anthocyanins standard curve was created using cyanidin-3-glucoside (C3G), ranging from 10 to 1000 µg/mL (y = 0.0002x − 0.1636, R^2^ = 0.9968).

The comparative data show no significant differences in total phenolic content, flavonoids, or anthocyanins between the purple waxy corn tassel extract and PPT. However, the EC50 values for DPPH, ABTS, and COX-II inhibition for PPT were significantly lower than those for the purple waxy corn tassel extract (*p* < 0.05, 0.05, and 0.001, respectively). No significant differences in antioxidant activity were observed between the two samples in the FRAP assay.

These results suggest that phytosome encapsulation enhances the biological activity of the purple waxy corn tassel extract. This improvement may be due to reduced degradation of active ingredients, likely resulting from the protective effects of the carrier wall, such as phosphatidylcholine used in the formulation.

### 3.2. Histopathological Changes of Lung Tissue

The morphological changes in lung tissue were examined and quantified, as shown in Figure 3. The lung tissues of the naïve control group, including bronchioles, alveoli, and pulmonary vessels, exhibited normal histological structures. However, in normal rats exposed to PM2.5-induced lung inflammation, a significant increase in pathological scores was observed, characterized by alveolar wall thickening, lung edema, and alveolar hemorrhage (*p* < 0.001 compared to the naïve control group).

Interestingly, all interventions in this study, including prednisolone at 1 mg/kg BW and PPT at doses of 100, 200, and 400 mg/kg BW, significantly reduced the pathological scores (*p* < 0.001, *p* < 0.05, *p* < 0.001, and *p* < 0.001, respectively, compared to the respiratory inflammation + vehicle group). These findings suggest that PPT effectively mitigates lung tissue damage induced by PM2.5.

### 3.3. Effect of PPT on Inflammatory Cytokines

Given the pivotal role of inflammation in lung damage, the effects of PPT on inflammatory markers, including COX-II activity, IL-1β, IL-6, IL-8, TNF-α, and NF-κB in lung tissue, were evaluated. As shown in Figure 4, rats with PM2.5-induced respiratory inflammation that received vehicle treatment exhibited a significant increase in COX-II activity (*p* < 0.01 compared to the naïve control group). However, this elevation was significantly mitigated by prednisolone and PPT at a dose of 400 mg/kg BW (both *p* < 0.05 compared to the respiratory inflammation + vehicle group).

Furthermore, PM2.5-exposed rats treated with vehicle also showed significant increases in IL-1β, IL-6, and IL-8 levels (all *p* < 0.001 compared to the naïve control group), as shown in Figure 5, Figure 6 and Figure 7. Treatment with prednisolone significantly reduced the expression of these cytokines (all *p* < 0.001 compared to the respiratory inflammation + vehicle group). Similarly, PPT at doses of 100, 200, and 400 mg/kg BW significantly reduced IL-1β levels (all *p* < 0.001 compared to the respiratory inflammation + vehicle group), IL-6 levels (*p* < 0.01, *p* < 0.01, and *p* < 0.001, respectively, compared to the respiratory inflammation + vehicle group), and IL-8 levels (*p* < 0.01, *p* < 0.01, and *p* < 0.001, respectively, compared to the respiratory inflammation + vehicle group).

Alterations in TNF-α and NF-κB levels in lung tissue were also investigated, with the results presented in Figure 8 and Figure 9. PM2.5-induced lung inflammation significantly increased both TNF-α and NF-κB levels in vehicle-treated rats (both *p* < 0.001 compared to the naïve control group). Interestingly, prednisolone and all doses of PPT significantly reduced the expression of TNF-α and NF-κB in the lungs of PM2.5-exposed rats (*p* < 0.001, *p* < 0.01, *p* < 0.001, and *p* < 0.001, respectively, compared to the respiratory inflammation + vehicle group).

### 3.4. Effect of PPT on Oxidative Stress Status in Lung Tissue

The effects of PPT on oxidative stress markers in lung tissue are summarized in Table 2. Exposure to PM2.5 significantly increased the MDA levels in the lung tissue of normal rats (*p* < 0.001 compared to the naïve control group). Only the group treated with PPT at a dose of 200 mg/kg BW showed a significant reduction in MDA levels in the lungs of rats with respiratory inflammation (*p* < 0.05 compared to the respiratory inflammation + vehicle group).

Additionally, the activities of key antioxidant enzymes, including SOD, CAT, and GSH-Px, were assessed. PM2.5 exposure significantly reduced the activities of SOD, CAT, and GSH-Px in normal rats (all *p* < 0.001 compared to the naïve control group). Treatment with prednisolone significantly increased SOD, CAT, and GSH-Px activities (all *p* < 0.001 compared to the respiratory inflammation + vehicle group). Moreover, PPT treatment at doses of 200 and 400 mg/kg BW significantly enhanced SOD and GSH-Px activities (*p* < 0.05 and *p* < 0.001, respectively, compared to the respiratory inflammation + vehicle group). In addition, PPT at 200 mg/kg BW significantly increased CAT activity (*p* < 0.001 compared to the respiratory inflammation + vehicle group).

These findings suggest that PPT helps mitigate oxidative stress in lung tissue by enhancing the activities of key antioxidant enzymes, contributing to its protective effects against PM2.5-induced damage.

### 3.5. Effect of PPT on Epigenetic Modifications in Lung Tissue

The role of epigenetic factors in lung inflammation was also investigated. As shown in Figure 10, PM2.5 inhalation significantly increased HDAC activity in lung tissue (*p* < 0.001 compared to the naïve control group). Treatment with PPT at doses of 200 and 400 mg/kg BW significantly decreased HDAC activity in the lung tissue of rats with PM2.5-induced lung inflammation (both *p* < 0.001 compared to the respiratory inflammation + vehicle group).

Additionally, the expression of sirtuins, a family of class III histone deacetylases involved in the regulation of cellular stress responses, was assessed. PM2.5 exposure led to a significant reduction in sirtuin expression in lung tissue (*p* < 0.001 compared to the naïve control group). This decrease was significantly reversed by treatment with prednisolone and all doses of PPT (*p* < 0.001, *p* < 0.01, and *p* < 0.001, respectively, compared to the respiratory inflammation + vehicle group), as shown in Figure 11.

These findings suggest that PPT exerts its protective effects against PM2.5-induced lung inflammation, at least in part, by modulating epigenetic factors such as HDAC and sirtuins, contributing to reduced inflammation and lung damage.

## 4. Discussion

Despite the recognized health benefits of phenolic compounds, flavonoids, and anthocyanins, their therapeutic potential remains limited due to challenges such as low bioavailability, rapid degradation, and poor stability, particularly when administered orally [35,36,37]. These limitations hinder their effectiveness in reaching target tissues in sufficient concentrations to exert their protective effects. However, encapsulation techniques, such as the phytosome delivery system employed in this study, have demonstrated the ability to overcome these barriers by enhancing the stability and bioavailability of these compounds, allowing for improved absorption and sustained therapeutic action [10,38]. In this study, PPT demonstrated enhanced bioactivity despite containing slightly lower levels of flavonoids and anthocyanins than the unencapsulated extract, which suggests that the phytosome formulation optimizes the delivery of bioactive compounds to target tissues.

The superior activity of PPT compared to the crude extract can be partly explained by the presence of other phenolic compounds, including phenolic acids and tannins, which also exhibit potent antioxidant and anti-inflammatory properties [39,40]. Although the total phenolic content in PPT was only slightly increased compared to the extract, this small increase could still contribute to the overall efficacy due to the combined effects of multiple phenolic compounds working synergistically. Furthermore, the phytosome encapsulation process may enhance the interaction of these phenolic compounds with cellular membranes, thereby facilitating their entry into cells and improving their bioactivity. This encapsulation technology not only improves the bioavailability of anthocyanins and other phenolic compounds but also protects them from degradation during digestion, ensuring that higher concentrations reach the lung tissue where they can exert their protective effects.

The findings of this study confirm that exposure to PM2.5 induces significant oxidative stress and inflammation in lung tissue, as evidenced by elevated levels of MDA, a marker of lipid peroxidation, and increased inflammatory markers, including COX-II, IL-1β, IL-6, IL-8, TNF-α, and NF-κB. These findings are consistent with previous research demonstrating that PM2.5 generates ROS, which in turn activate pro-inflammatory signaling pathways [41,42]. The upregulation of NF-κB, a transcription factor that controls the expression of many inflammatory genes, suggests that oxidative stress and inflammation are tightly interconnected in the pathology of PM2.5-induced lung damage [43,44]. Importantly, PPT significantly reduced these markers, indicating its ability to break the vicious cycle of oxidative stress and inflammation.

Interestingly, all doses of PPT were effective in reducing pro-inflammatory cytokines, although the suppression of COX-II was only observed at the highest dose (400 mg/kg), while the reduction of MDA levels was significant at the medium dose (200 mg/kg). This suggests that different bioactive components within PPT may target specific aspects of the inflammatory response and oxidative stress, with higher doses required to achieve the pharmacological threshold needed to suppress COX-II activity. COX-II is a key enzyme in the inflammatory cascade, catalyzing the conversion of arachidonic acid to prostaglandins, which mediate pain and inflammation. The selective suppression of COX-II at the highest dose may indicate that PPT acts on this enzyme through dose-dependent mechanisms, potentially requiring higher concentrations of anthocyanins or other active compounds to exert its full effect.

The failure of the highest dose to reduce MDA levels may be attributed to its lesser effect on CAT activity. CAT is one of the main scavenger enzymes responsible for neutralizing hydrogen peroxide, a ROS that contributes to oxidative damage. Although PPT increased the activity of SOD, CAT, and GSH-Px overall, the specific response of CAT to the high dose was less pronounced, possibly explaining the incomplete reduction in oxidative stress markers like MDA. This suggests that the antioxidant response may be more complex, with different enzymes responding variably to PPT treatment depending on the dose.

Moreover, the role of epigenetic mechanisms in the protective effects of PPT is of particular interest. HDAC and sirtuins (class III HDAC) are key regulators of gene expression, controlling the balance between pro-inflammatory and anti-inflammatory signals at the epigenetic level. HDAC3, a class I HDAC, has been implicated in promoting inflammation and fibrosis in various solid organs, including the lungs, by deacetylating histones and other proteins involved in inflammatory gene transcription [45]. In contrast, sirtuins, especially SIRT1, play a protective role by deacetylating NF-κB, thereby suppressing the transcription of pro-inflammatory cytokines and attenuating the inflammatory response [46,47]. Our data demonstrate that PM2.5 exposure upregulates HDAC expression while downregulating sirtuins, contributing to an exaggerated inflammatory response. Notably, PPT treatment reversed these changes, with all doses reducing HDAC expression and increasing sirtuin levels. This suggests that PPT exerts its anti-inflammatory effects not only through direct inhibition of inflammatory mediators but also through epigenetic modulation, particularly by enhancing the activity of sirtuins and inhibiting HDAC.

The modulation of HDAC and sirtuins by PPT could represent a key mechanism underlying its multifaceted protective effects. By promoting SIRT1 activity, PPT may enhance the deacetylation of NF-κB, leading to reduced transcription of inflammatory cytokines such as TNF-α, IL-6, and IL-8. This is consistent with the observed decrease in these cytokines across all treatment groups. Additionally, the inhibition of HDAC3 by PPT may prevent the deacetylation of histones and other proteins involved in pro-fibrotic and pro-inflammatory pathways, further contributing to the attenuation of lung damage [48,49]. These findings suggest that the anti-inflammatory and antioxidant effects of PPT are closely linked to its ability to modulate epigenetic regulators, adding an additional layer of complexity to its mechanism of action.

The absence of a clear dose-dependent response in the current study may be due to the complex interactions between the various bioactive compounds in PPT and their differential effects on distinct molecular targets. Phytosome formulations contain a variety of compounds that may interact with each other, potentially masking dose-dependent effects. Furthermore, the protective effects of PPT may be mediated, at least in part, by epigenetic mechanisms that do not necessarily follow a linear relationship with dose. Anthocyanins, which have been shown to modulate HDAC and sirtuin activity, likely play a role in this epigenetic regulation. Therefore, the protective effects of PPT may involve both direct biochemical actions on oxidative stress and inflammation, as well as indirect effects through the modulation of gene expression at the epigenetic level.

In summary, this study demonstrates that PPT provides protection against PM2.5-induced lung damage through various mechanisms, including the suppression of inflammatory cytokines, enhancement of antioxidant defenses, and modulation of epigenetic regulators such as HDAC and sirtuins. These findings highlight the potential of PPT as a complementary therapeutic agent for managing respiratory inflammation, particularly in the context of environmental pollution. Future research should aim to clarify the specific interactions between the bioactive compounds in PPT and their molecular targets, as well as investigate the long-term effects of PPT in chronic exposure models to better understand its role as a functional food ingredient.

## 5. Conclusions

This study provides novel insights into the protective effects of PPT against PM2.5-induced lung inflammation. Our findings indicate that these protective effects are primarily mediated through epigenetic modifications, specifically the regulation of HDAC and sirtuins, which effectively suppress pro-inflammatory cytokines and mitigate lung injury. Furthermore, PPT’s modulation of COX-II and its antioxidant properties enhance its ability to counteract inflammation and oxidative stress, with effects influenced by dosage. Given these promising results, PPT shows potential as a functional food ingredient to support the management of lung inflammation and injury linked to PM2.5 exposure. Future clinical studies are essential to validate these protective effects and explore the applicability of PPT in human populations, further emphasizing the importance of integrating functional food ingredients into health-promoting strategies.

## Figures and Tables

**Figure 1 foods-13-03258-f001:**
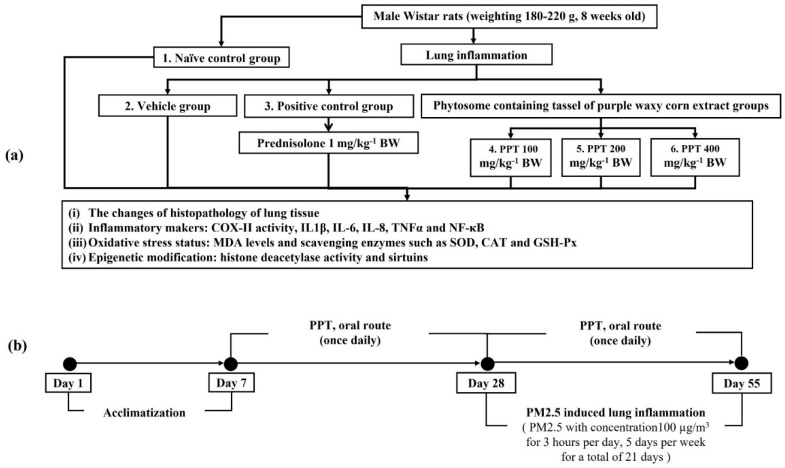
Schematic diagram illustrating the experimental procedures. (**a**) Experimental protocol for PPT treatment and the assessment of various parameters. (**b**) Timeline for PM2.5-induced lung inflammation and PPT treatment administration. COX-II: cyclooxygenase-2; IL-1β: interleukin-1 beta; IL-6: interleukin-6; IL-8: interleukin-8; TNF-α: tumor necrosis factor-α; NF-κB: nuclear factor-kappa B; PPT100, PPT200, and PPT400: phytosome containing purple waxy corn tassel extract at doses of 100, 200, and 400 mg/kg BW, respectively.

**Figure 2 foods-13-03258-f002:**
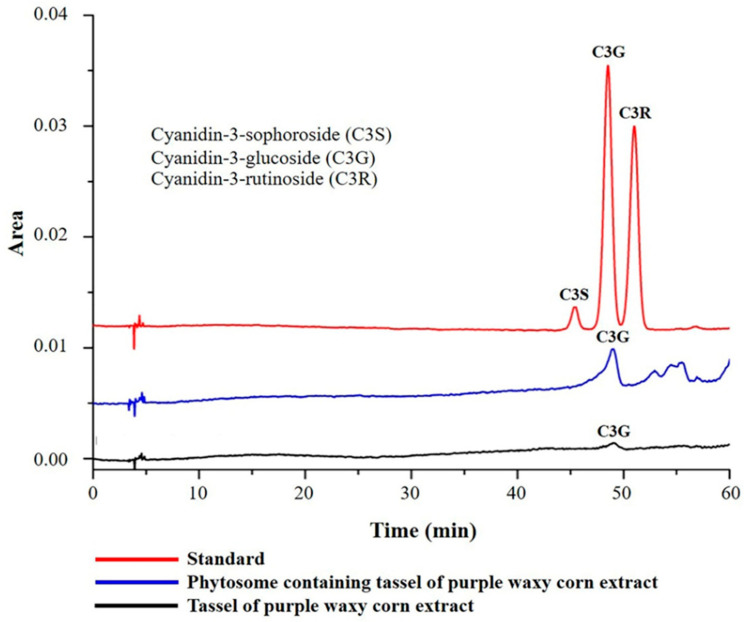
High-performance liquid chromatography (HPLC) profile used to identify and quantify anthocyanin content.

**Figure 3 foods-13-03258-f003:**
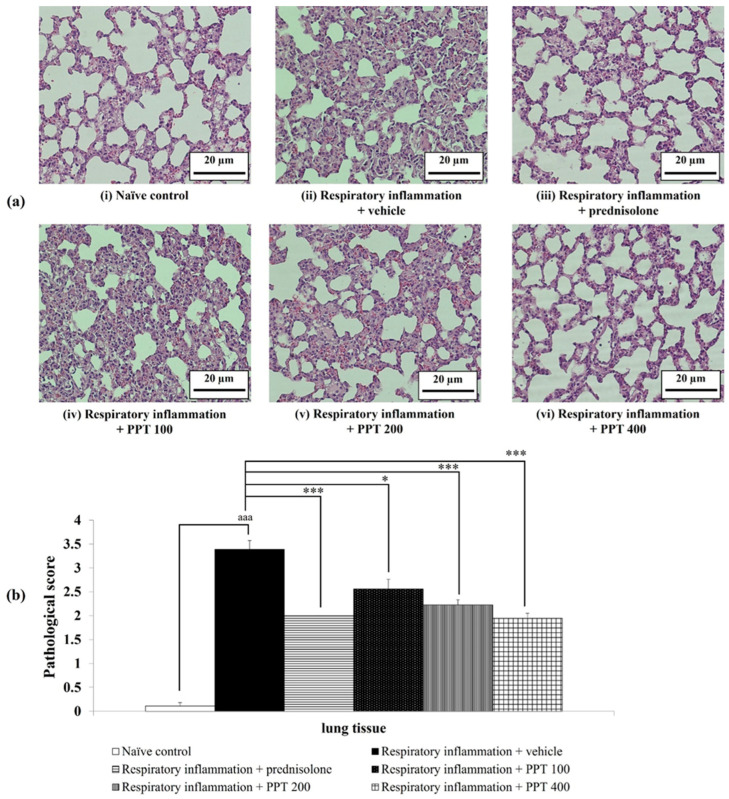
Effect of various doses of PPT on pathological changes in lung tissue induced by PM2.5. (**a**) Light microscopy images of lung tissue stained with hematoxylin and eosin (H&E) at 40× magnification. (**b**) Quantified scoring of pathological changes in lung tissue. Data are presented as mean ± SEM (*n* = 6/group). ^aaa^ *p* < 0.001; comparison between naïve control rats receiving vehicle. *, *** *p* < 0.05 and 0.001, respectively; comparison to respiratory inflammation rats receiving PM2.5 and vehicle. Prednisolone: administered at a dose of 1 mg/kg BW; PPT100, PPT200, and PPT400: phytosome containing purple waxy corn tassel extract at doses of 100, 200, and 400 mg/kg BW, respectively.

**Figure 4 foods-13-03258-f004:**
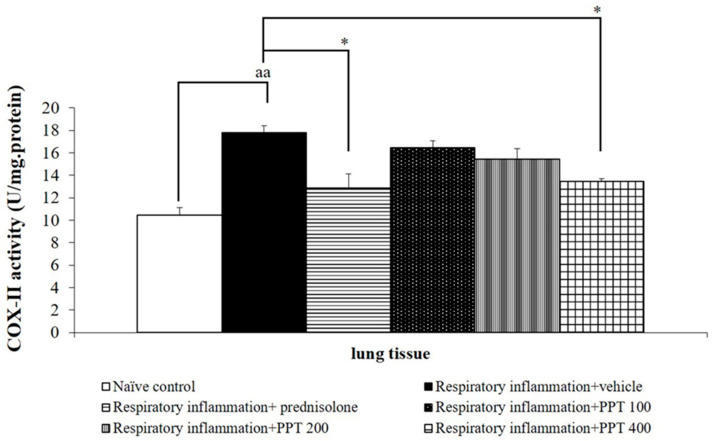
Effect of various doses of PPT on COX-II activity in lung tissue. Data are presented as mean ± SEM (*n* = 6/group). ^aa^ *p* < 0.01; comparison between naïve control rats receiving vehicle. * *p* < 0.05; comparison to respiratory inflammation rats receiving PM2.5 and vehicle. Prednisolone: administered at a dose of 1 mg/kg BW; PPT100, PPT200, and PPT400: phytosome containing purple waxy corn tassel extract at doses of 100, 200, and 400 mg/kg BW, respectively.

**Figure 5 foods-13-03258-f005:**
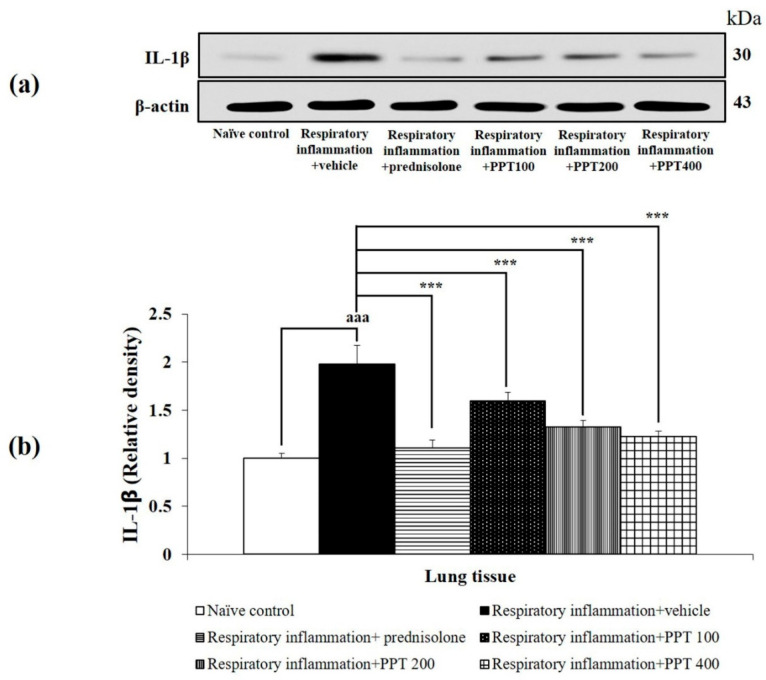
Effect of various doses of PPT on IL-1β expression in lung tissue. (**a**) Representative Western blot showing IL-1β levels. (**b**) Quantification of relative IL-1β density. Data are presented as mean ± SEM (*n* = 6/group). ^aaa^ *p* < 0.001; comparison between naïve control rats receiving vehicle. *** *p* < 0.001; comparison to respiratory inflammation rats receiving PM2.5 and vehicle. Prednisolone: administered at a dose of 1 mg/kg BW; PPT100, PPT200, and PPT400: phytosome containing purple waxy corn tassel extract at doses of 100, 200, and 400 mg/kg BW, respectively.

**Figure 6 foods-13-03258-f006:**
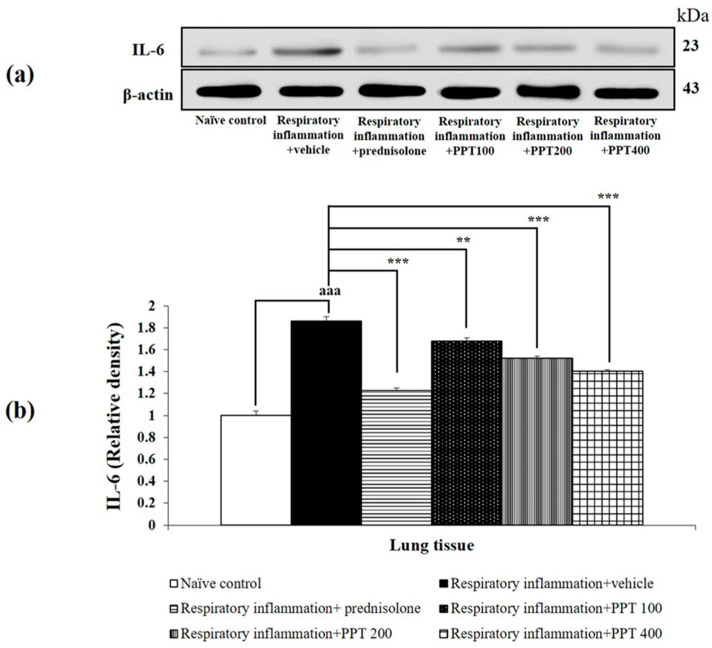
Effect of various doses of PPT on IL-6 expression in lung tissue. (**a**) Representative Western blot showing IL-6 levels. (**b**) Quantification of relative IL-6 density. Data are presented as mean ± SEM (*n* = 6/group). ^aaa^ *p* < 0.001; compared to naïve control rats receiving vehicle. **, *** *p* < 0.01 and 0.001, respectively; compared to respiratory inflammation rats receiving PM2.5 and vehicle. Prednisolone: administered at a dose of 1 mg/kg BW; PPT100, PPT200, and PPT400: phytosome containing purple waxy corn tassel extract at doses of 100, 200, and 400 mg/kg BW, respectively.

**Figure 7 foods-13-03258-f007:**
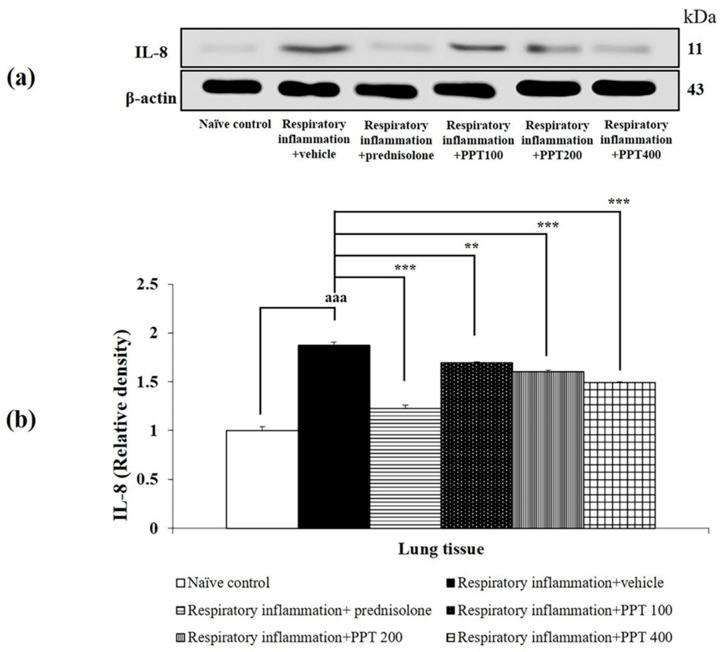
Effect of various doses of PPT on IL-8 expression in lung tissue. (**a**) Representative Western blot showing IL-8 levels. (**b**) Quantification of relative IL-8 density. Data are presented as mean ± SEM (*n* = 6/group). ^aaa^ *p* < 0.001; compared to naïve control rats receiving vehicle. **, *** *p* < 0.01 and 0.001, respectively; compared to respiratory inflammation rats receiving PM2.5 and vehicle. Prednisolone: administered at a dose of 1 mg/kg BW; PPT100, PPT200, and PPT400: phytosome containing purple waxy corn tassel extract at doses of 100, 200, and 400 mg/kg BW, respectively.

**Figure 8 foods-13-03258-f008:**
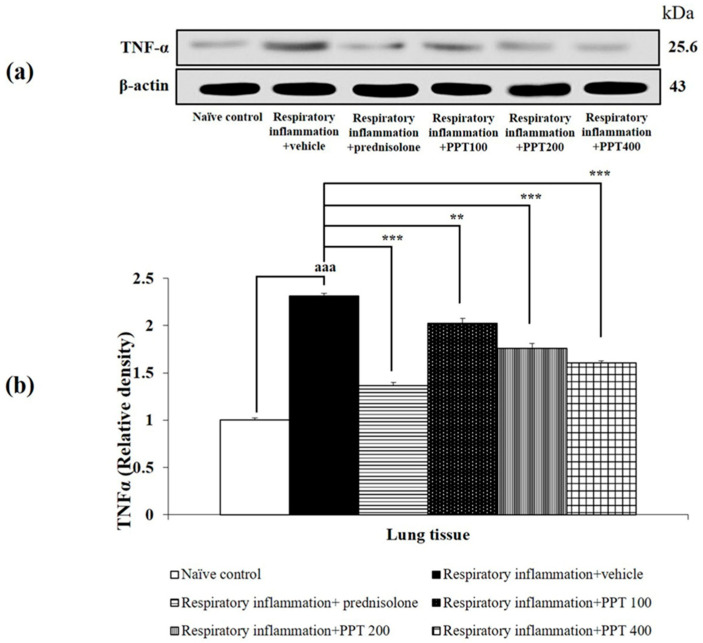
Effect of various doses of PPT on TNF-α expression in lung tissue. (**a**) Representative Western blot showing TNF-α levels. (**b**) Quantification of relative TNF-α density. Data are presented as mean ± SEM (*n* = 6/group). ^aaa^ *p* < 0.001; compared to naïve control rats receiving vehicle. **, *** *p* < 0.01 and 0.001, respectively; compared to respiratory inflammation rats receiving PM2.5 and vehicle. Prednisolone: administered at a dose of 1 mg/kg BW; PPT100, PPT200, and PPT400: phytosome containing purple waxy corn tassel extract at doses of 100, 200, and 400 mg/kg BW, respectively.

**Figure 9 foods-13-03258-f009:**
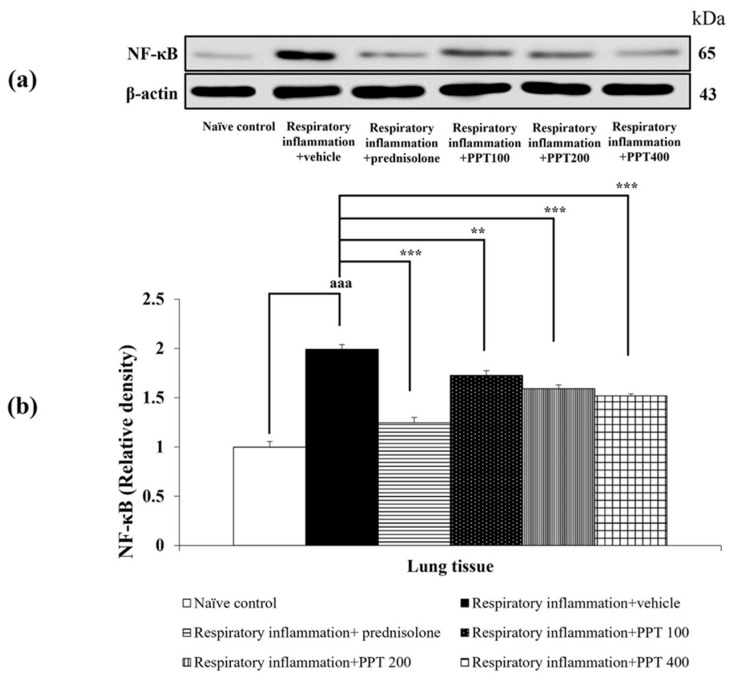
Effect of various doses of PPT on NF-κB expression in lung tissue. (**a**) Representative Western blot showing NF-κB levels. (**b**) Quantification of relative NF-κB density. Data are presented as mean ± SEM (*n* = 6/group). ^aaa^ *p* < 0.001; compared to naïve control rats receiving vehicle. **, *** *p* < 0.01 and 0.001, respectively; compared to respiratory inflammation rats receiving PM2.5 and vehicle. Prednisolone: administered at a dose of 1 mg/kg BW; PPT100, PPT200, and PPT400: phytosome containing purple waxy corn tassel extract at doses of 100, 200, and 400 mg/kg BW, respectively.

**Figure 10 foods-13-03258-f010:**
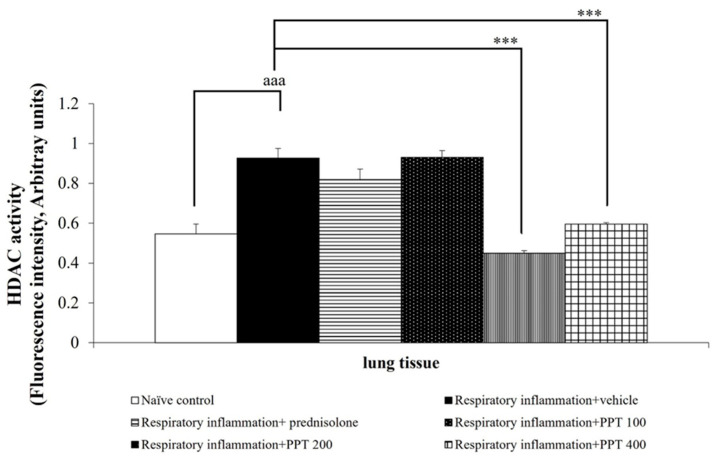
Effect of various doses of PPT on HDAC activity expression in lung tissue. Data are presented as mean ± SEM (*n* = 6/group). ^aaa^ *p* < 0.001; compared to naïve control rats receiving vehicle. *** *p* < 0.001; compared to respiratory inflammation rats receiving PM2.5 and vehicle. Prednisolone: administered at a dose of 1 mg/kg BW; PPT100, PPT200, and PPT400: phytosome containing purple waxy corn tassel extract at doses of 100, 200, and 400 mg/kg BW, respectively.

**Figure 11 foods-13-03258-f011:**
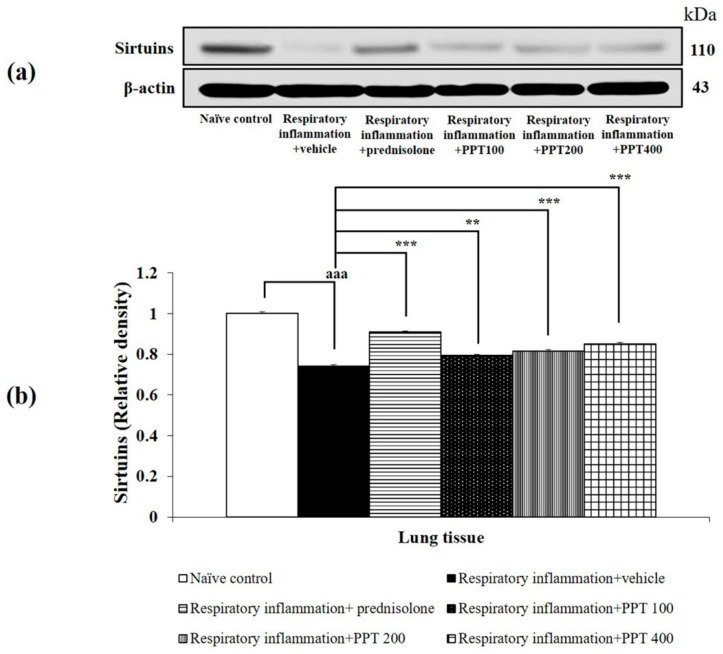
Effect of various doses of PPT on sirtuins expression in lung tissue. (**a**) Representative Western blot showing sirtuins levels. (**b**) Quantification of relative sirtuins density. Data are presented as mean ± SEM (*n* = 6/group). ^aaa^ *p* < 0.001; compared to naïve control rats receiving vehicle. **, *** *p* < 0.01 and 0.001, respectively; compared to respiratory inflammation rats receiving PM2.5 and vehicle. Prednisolone: administered at a dose of 1 mg/kg BW; PPT100, PPT200, and PPT400: phytosome containing purple waxy corn tassel extract at doses of 100, 200, and 400 mg/kg BW, respectively.

**Table 1 foods-13-03258-t001:** Phenolic compositions and biological activities of the tassel of purple waxy corn extract and PPT.

Parameters	Units	Tassel of Purple Waxy Corn Extract	PPT	Standard Reference
Active compounds				
Total phenolic content	mg Gallic acid/mg	40.257 ± 0.024	41.819 ± 0.259	-
Total flavonoids content	mg Quercetin/mg	5.938 ± 0.006	5.749 ± 0.019	-
Total anthocyanins content	mg C3G /mg	8.921 ± 0.056	8.780 ± 0.028	-
Antioxidant activities				
DPPH	EC50 (mg/mL)	1.062 ± 0.006	0.685 ± 0.010 *	0.039 ± 0.001, Trolox
FRAP	EC50 (mg/mL)	0.549 ± 0.011	0.896 ± 0.002	Trolox
ABTS	EC50 (mg/mL)	1.729 ± 0.022	1.240 ± 0.003 *	1.479 ± 0.029, Trolox
Inflammatory marker				
COX-II	EC50 (mg/mL)	3.098 ± 0.250	2.261 ± 0.19 ***	2.218 ± 0.041, Indomethacin

Data are presented as mean ± SEM. *, *** *p* < 0.05 and 0.001, respectively; comparisons are made between the tassel of purple waxy corn extract and PPT.

**Table 2 foods-13-03258-t002:** Effect of various doses of PPT on oxidative stress markers in lung tissue.

Treatment Groups	MDA (ng/mg Protein)	CAT (units/mg Protein)	SOD (units/mg Protein)	GSH-Px (units/mg Protein)
Naïve control	0.107 ± 0.004	20.140 ± 1.565	12.138 ± 0.592	8.092 ± 0.547
Respiratory inflammation + vehicle	0.223 ± 0.006 ^aaa^	9.627 ± 0.238 ^aaa^	4.729 ± 0.499 ^aaa^	4.648 ± 0.170 ^aaa^
Respiratory inflammation + prednisolone	0.201 ± 0.006	18.172 ± 1.234 ***	10.454 ± 0.510 ***	7.938 ± 0.257 ***
Respiratory inflammation + PPT 100	0.208 ± 0.007	11.281 ± 0.945	5.443 ± 0.293	4.840 ± 0.405
Respiratory inflammation + PPT 200	0.194 ± 0.012 *	19.038 ± 1.030 ***	7.338 ± 0.306 *	6.176 ± 0.350 *
Respiratory inflammation + PPT 400	0.220 ± 0.009	12.310 ± 1.147	9.060 ± 1.365 ***	7.725 ± 0.194 ***

Data are presented as mean ± SEM (n = 6/group). ^aaa^
*p* < 0.001; comparisons are made between naïve control rats that received vehicle and respiratory inflammation rats that received PM2.5 and vehicle, with *, *** *p* < 0.05 and 0.001, respectively. Prednisolone was administered at a dose of 1 mg/kg BW; PPT 100, PPT 200, and PPT 400 represent the phytosome containing tassel of purple waxy corn extract at doses of 100, 200, and 400 mg/kg BW, respectively.

## Data Availability

The data presented in this study are available on request from the corresponding author due to all data are in the process of petty patent registration.

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
