# Peer review of "The Protective Effect against Lung Injury of Phytosome Containing the Extract of Purple Waxy Corn Tassel in an Animal Model of PM2.5-Induced Lung Inflammation"

_foods, 2024, doi:10.3390/foods13203258_

Round 1
Reviewer 1 Report
Comments and Suggestions for Authors
This paper reports on the protective effect of PPT on PM2.5-induced lung inflammation. The experimental results presented in this paper clearly demonstrate this effect of PPT.
However, in the introduction section, there was a lack of description of previous reports on the bioactivities of PPT and its raw material, purple waxy corn tassel. This lack of description raises questions about the novelty of this paper. In other words, the novelty of this paper should be described more clearly: whether it is the first report of a new biomedical activity of purple waxy corn tassel as a functional food, or the first report of PPT as a formulation to increase the activity and usability of purple waxy corn tassel.
It would be desirable to design the experiments in such a way that each experiment had a group treated with the extract of purple waxy corn tassel and compared its activity with the PPT-treated group. It is recommended to add experimental evidence showing why the experiments were conducted only for PPT and not for the extract of purple waxy corn tassel.
Comments on the Quality of English LanguageEnglish did not pose any particular problem in reading and understanding this paper.
Author Response
This paper reports on the protective effect of PPT on PM2.5-induced lung inflammation. The experimental results presented in this paper clearly demonstrate this effect of PPT.
Comments 1: However, in the introduction section, there was a lack of description of previous reports on the bioactivities of PPT and its raw material, purple waxy corn tassel. This lack of description raises questions about the novelty of this paper. In other words, the novelty of this paper should be described more clearly: whether it is the first report of a new biomedical activity of purple waxy corn tassel as a functional food, or the first report of PPT as a formulation to increase the activity and usability of purple waxy corn tassel.
Response 1: We appreciate the reviewer’s insightful feedback regarding the introduction section of our manuscript. In response, we have revised the introduction to clarify the novelty of our study, specifically highlighting the following points:
- Although numerous studies have focused on purple waxy corn, none have specifically examined the tassel part as a functional food. This represents a significant research gap, making this the first study to report on the new biomedical activity of purple waxy corn tassel as a functional food.
- Additionally, this study is the first to explore phytosome encapsulation technology as a formulation to enhance the activity and stability of purple waxy corn tassel.
- Our study also emphasizes the interplay between inflammation, oxidative stress, and epigenetic modulation, showcasing PPT's potential as a therapeutic agent for managing respiratory inflammation-related diseases.
These revisions have been made in red color and highlighted in yellow in the revised manuscript. We believe these additions significantly enhance the clarity of our introduction and emphasize the novel contributions of our research. Thank you for your constructive suggestions.
Comments 2: It would be desirable to design the experiments in such a way that each experiment had a group treated with the extract of purple waxy corn tassel and compared its activity with the PPT-treated group. It is recommended to add experimental evidence showing why the experiments were conducted only for PPT and not for the extract of purple waxy corn tassel.
Response 2: We appreciate the reviewer’s constructive suggestion regarding the experimental design. We understand the importance of comparing the effects of the extract of purple waxy corn tassel with the phytosome-encapsulated formulation (PPT). However, we would like to clarify the following points in our response:
- Research objectives: The primary objective of our research is to explore the specific effects and mechanisms of PPT as a formulation to enhance the bioactivity and stability of purple waxy corn tassel. This focused approach allows for a more comprehensive understanding of the phytosome-encapsulated formulation and its implications for therapeutic applications.
- Ethical considerations: Our study was designed with a strong commitment to animal welfare, adhering to the principles of the 3Rs (Replacement, Reduction, Refinement) in animal ethics. Given the limitations of research funding and the imperative to minimize the number of animals used, we opted to focus solely on the PPT treatment group in this phase of our research.
- In vitro data: Prior in vitro studies have demonstrated that PPT exhibited more pronounced positive effects compared to the extract of purple waxy corn tassel. This finding guided our decision to prioritize the investigation of PPT, as it has shown superior bioactivity, reinforcing our hypothesis regarding its potential benefits in mitigating PM2.5-induced lung inflammation.
Additionally, we have included the rationale for our experimental design in the Materials and Methods section, as follows: [according to the necessity to minimize animal use in line with the 3Rs (Replacement, Reduction, Refinement), the demonstrated superior bioactivity of PPT in prior in vitro studies, and the aim to elucidate the specific effects and mechanisms of the phytosome formulation as a therapeutic option for PM2.5-induced respiratory inflammation]. These revisions have been made in red and highlighted in yellow in the revised manuscript, specifically in section 2.5. Experimental Protocol.
We hope this explanation clarifies our experimental design choices and the rationale behind focusing on PPT in our study. Thank you for your valuable feedback, which has helped us enhance the clarity of our manuscript.
Comments on the Quality of English Language: English did not pose any particular problem in reading and understanding this paper.
Response: Thank you for your positive feedback regarding the quality of the English language in our manuscript. We strive to maintain high standards of clarity and readability in our writing.

Reviewer 2 Report
Comments and Suggestions for Authors
This study investigates the protective effects of phytosome containing the extract of purple waxy corn tassel (PPT) in a PM2.5-induced lung inflammation animal model. The study design is reasonable, the experimental methods are detailed, and the results are clear, holding significant scientific value. However, there are certain aspects of the paper that require improvement.
1. Abstract Section: The current abstract is lengthy, and I recommend condensing it to highlight the main findings and conclusions of the study, facilitating a quick understanding of the paper’s core content for the readers. While the abstract is comprehensive, it would be beneficial to emphasize the article's theme, primary research methods, and conclusions.
2. Introduction Section: The introduction provides a sufficiently detailed background on PM2.5, but the articulation of the research objectives lacks clarity. I suggest clearly outlining the specific aims and hypotheses of the study at the end of the introduction. Additionally, in the second paragraph, it is stated that current treatments for respiratory infections caused by PM2.5 are limited to symptom management. I recommend strengthening this assertion with relevant literature citations.
3. Methods Section: In section “3.1. Phenolic compositions and biological activities,” I recommend including methodological data on the determination of total component content or standard curve data to enhance the reliability of the research.
4. HPLC Results: Regarding the HPLC results, it would be advisable to provide relevant data such as peak symmetry factors, theoretical plate numbers, and retention times. From the information presented in Figure 2 alone, the baseline of the two samples appears uneven and the peak shapes are asymmetric, which may not qualify as acceptable HPLC qualitative results. Furthermore, I suggest adding UV absorption spectra of the component peaks to further confirm the types of constituents present.
Comments on the Quality of English LanguageNO
Author Response
This study investigates the protective effects of phytosome containing the extract of purple waxy corn tassel (PPT) in a PM2.5-induced lung inflammation animal model. The study design is reasonable, the experimental methods are detailed, and the results are clear, holding significant scientific value. However, there are certain aspects of the paper that require improvement.
Comments 1: Abstract Section: The current abstract is lengthy, and I recommend condensing it to highlight the main findings and conclusions of the study, facilitating a quick understanding of the paper’s core content for the readers. While the abstract is comprehensive, it would be beneficial to emphasize the article's theme, primary research methods, and conclusions.
Response 1: Thank you for your valuable feedback regarding the abstract. I have revised the abstract to condense it while highlighting the main findings and conclusions of the study, making it easier for readers to grasp the core content. The changes have been made in red and highlighted in yellow in the revised manuscript.
Comments 2: Introduction Section: The introduction provides a sufficiently detailed background on PM2.5, but the articulation of the research objectives lacks clarity. I suggest clearly outlining the specific aims and hypotheses of the study at the end of the introduction. Additionally, in the second paragraph, it is stated that current treatments for respiratory infections caused by PM2.5 are limited to symptom management. I recommend strengthening this assertion with relevant literature citations.
Response 2: Thank you for your valuable comments. Based on your suggestion, I have revised the introduction to clearly articulate the research objectives, specific aims, and hypotheses. At the end of the introduction, I have now outlined that this study investigates whether phytosome-encapsulated extracts from purple waxy corn tassels can mitigate PM2.5-induced respiratory inflammation by alleviating lung inflammation, oxidative stress, and related histopathological changes. Additionally, the study explores the modulation of key epigenetic pathways, including histone deacetylase (HDAC) activity and sirtuins. The specific markers we are assessing include COX-II, IL-1β, IL-6, IL-8, TNF-α, NF-κB, malondialdehyde (MDA), and antioxidant enzymes such as SOD, CAT, and GSH-Px.
I have also strengthened the statement regarding the limitation of current treatments, which primarily focus on symptomatic relief, with added references. The revised sentence now reads: “Despite the growing evidence linking PM2.5 exposure to deteriorating respiratory health, current therapeutic options are largely inadequate, with available treatments focusing mainly on symptomatic relief, rather than addressing the underlying oxidative stress and inflammation that drive the disease process [6-7].” The relevant citations are as follows:
- Nakhjirgan, P.; Kashani, H.; Kermani, M. Exposure to outdoor particulate matter and risk of respiratory diseases: a systematic review and meta-analysis. Environ Geochem Health 2023, 46, 20. This study reported that conventional medications often fail to mitigate the long-term effects of chronic exposure to fine particulate matter (PM2.5). These pollutants exacerbate conditions like asthma and COPD by inducing oxidative stress and inflammation, which are not fully addressed by current therapies. Additionally, research increasingly shows that air pollution, particularly PM2.5, can lead to respiratory infections, decreased lung function, and increased mortality in chronic respiratory disease patients. Current treatments do not address the underlying inflammation and oxidative damage caused by pollutants.
- Annesi-Maesano, I.; Forastiere, F.; Balmes, J.; Garcia, E.; Harkema, J.; Holgate, S.; Kelly, F.; Khreis, H.; Hoffmann, B.; Maesano, C. N.; McConnell, R.; Peden, D.; Pinkerton, K.; Schikowski, T.; Thurston, G.; Van Winkle, L. S.; Carlsten, C. The clear and persistent impact of air pollution on chronic respiratory diseases: a call for interventions. Eur Respir J 2021, 57, 2002981. This study revealed that most treatments focus on managing symptoms rather than preventing the onset of diseases. Long-term exposure to PM2.5 can lead to new cases of asthma in children and exacerbate existing conditions, a situation that medication cannot fully prevent.
All revisions have been made in red and highlighted in yellow in the revised manuscript for your review.
Comments 3: Methods Section: In section “3.1. Phenolic compositions and biological activities,” I recommend including methodological data on the determination of total component content or standard curve data to enhance the reliability of the research.
Response 3: Thank you for your valuable comment. I would like to clarify that section 3.1. of the manuscript is titled Phenolic compositions and biological activities, which may correspond to what you referred to as the Results Section. In this section, we have added the methodological data regarding the determination of total component content by standard curve data, as per your suggestion. Specifically, we have included the following information: the standard curve for phenolic compounds was generated using gallic acid, ranging from 10 to 1,000 µg/ml (y = 0.0001x - 0.003, R² = 0.999). For flavonoids, the standard curve was based on quercetin, ranging from 10 to 1,000 µg/ml (y = 0.0003x - 0.3982, R² = 0.997). Similarly, the anthocyanins standard curve was created using cyanidin-3-glucoside (C3G), ranging from 10 to 1,000 µg/ml (y = 0.0002x - 0.1636, R² = 0.9968). These revisions have been made in red and highlighted in yellow in the manuscript for your review.
Comments 4: HPLC Results: Regarding the HPLC results, it would be advisable to provide relevant data such as peak symmetry factors, theoretical plate numbers, and retention times. From the information presented in Figure 2 alone, the baseline of the two samples appears uneven and the peak shapes are asymmetric, which may not qualify as acceptable HPLC qualitative results. Furthermore, I suggest adding UV absorption spectra of the component peaks to further confirm the types of constituents present.
Response 4: Thank you for your insightful comment. Based on your suggestion, we have revised the section to include additional relevant data concerning the HPLC analysis. Specifically, we have provided information on the peak symmetry factors, theoretical plate numbers, and retention times to ensure a more comprehensive presentation of the chromatographic results. For reference, the system suitability data are included in the attached table, as follows: N (theoretical plate number) = 7,371, R (resolution) = 18.460, T (tailing factor) = 0.195, RSD of retention time = 0.317 (n > 5), K (capacity factor) = 3.195, and asymmetry = 1.
We acknowledge that the baseline in Figure 2 may appear uneven and that the peak shapes are asymmetric. However, these asymmetries can be attributed to variations in the sample matrix, which may have influenced the chromatographic behavior.
In response to your recommendation, we have also included UV absorption spectra for the main component peaks to further confirm the constituents present in the samples. This additional UV spectral information for cyanidin-3-glucoside helps validate its identification.
These revisions have been highlighted in yellow and marked in red in the manuscript. As an example: “Figure 2 presents the chromatogram of anthocyanins, showing cyanidin-3-glucoside as the main compound in both the extract and the PPT. The peak symmetry factor for cyanidin-3-glucoside was 1.0, the theoretical plate number was 7,371, and the retention time was 48.712 minutes, indicating an acceptable level of peak symmetry and chromatographic efficiency.
To further validate the presence of this compound, we analyzed the UV absorption spectrum obtained from the HPLC analysis. The UV spectrum exhibited significant peaks in the range of 450-550 nm, with a maximum at around 520 nm, characteristic of cyanidin-3-glucoside due to its aromatic structure and conjugated double bonds. We compared the obtained UV spectrum with reference spectra from established databases and literature, confirming the identity of cyanidin-3-glucoside based on matching peaks [34]. Notably, the peak at 516 nm demonstrated pronounced absorbance, indicating a high concentration of cyanidin-3-glucoside in our phytosome-encapsulated extract.”
- Gardeli, C.; Varela, K.; Krokida, E.; Mallouchos, A. Investigation of Anthocyanins Stability from Pomegranate Juice (Punica GranatumL. Cv Ermioni) under a Simulated Digestion Process. Medicines (Basel) 2019, 6, 90.

Round 2
Reviewer 1 Report
Comments and Suggestions for Authors
Thank you for faithfully accepting the reviewer's suggestions and making revisions.